ecology, health and disease and epidemiology, molecular biology

aquaculture, infectious disease, disease spillover, wild salmon, microparasites

**Author for correspondence:**
Dylan Shea
e-mail: dylan.shea@mail.utoronto.ca

†Shared last authors.

# Environmental DNA from multiple pathogens is elevated near active Atlantic salmon farms

Dylan Shea[1], Andrew Bateman[1,2,3], Shaorong Li[4], Amy Tabata[4], Angela Schulze[4], Gideon Mordecai[5], Lindsey Ogston[1], John P. Volpe[6], L. Neil Frazer[7], Brendan Connors[8], Kristina M. Miller[4], Steven Short[1,9,†] and Martin Krkošek[1,2,†]

[1]Department of Ecology and Evolutionary Biology, University of Toronto, Ontario, Canada
[2]Salmon Coast Field Station, Simoom Sound, British Columbia, Canada
[3]Pacific Salmon Foundation, Vancouver, British Columbia, Canada
[4]Pacific Biological Station, Fisheries and Oceans Canada, Nanaimo, British Columbia, Canada
[5]Department of Medicine, University of British Columbia, Vancouver, British Columbia, Canada
[6]School of Environmental Studies, University of Victoria, Victoria, British Columbia, Canada
[7]Department of Earth Sciences, University of Hawaii at Mānoa, Honolulu, Hawaii, Canada
[8]Institute of Ocean Sciences, Fisheries and Oceans Canada, Sidney, British Columbia, Canada
[9]Department of Biology, University of Toronto Mississauga, Mississauga, British Columbia, Canada

DS, 0000-0001-6524-5381; SS, 0000-0002-6576-4883; MK, 0000-0001-7591-7954

The spread of infection from reservoir host populations is a key mechanism for disease emergence and extinction risk and is a management concern for salmon aquaculture and fisheries. Using a quantitative environmental DNA methodology, we assessed pathogen environmental DNA in relation to salmon farms in coastal British Columbia, Canada, by testing for 39 species of salmon pathogens (viral, bacterial, and eukaryotic) in 134 marine environmental samples at 58 salmon farm sites (both active and inactive) over 3 years. Environmental DNA from 22 pathogen species was detected 496 times and species varied in their occurrence among years and sites, likely reflecting variation in environmental factors, other native host species, and strength of association with domesticated Atlantic salmon. Overall, we found that the probability of detecting pathogen environmental DNA (eDNA) was 2.72 (95% CI: 1.48, 5.02) times higher at active versus inactive salmon farm sites and 1.76 (95% CI: 1.28, 2.42) times higher per standard deviation increase in domesticated Atlantic salmon eDNA concentration at a site. If the distribution of pathogen eDNA accurately reflects the distribution of viable pathogens, our findings suggest that salmon farms serve as a potential reservoir for a number of infectious agents; thereby elevating the risk of exposure for wild salmon and other fish species that share the marine environment.

## 1. Introduction

In multi-host infectious disease systems, reservoir host species are those that alone can maintain the parasite and sustain transmission to other host species [1]. Transmission from reservoir hosts is implicated in the emergence or re-emergence of infectious diseases [2], and is the primary mechanism by which disease can elevate extinction risk of wildlife [3]. In the marine environment, low barriers to parasite dispersal and large migrations of many marine fauna can facilitate multi-host parasite transmission [4,5]. Fishing and aquaculture can deplete wild fish populations or introduce new reservoir host populations that can alter multi-host transmission dynamics [6,7]. However, despite the importance of reservoir hosts for parasite spread, disease emergence, and

Proc. R. Soc. B 287: 20202010

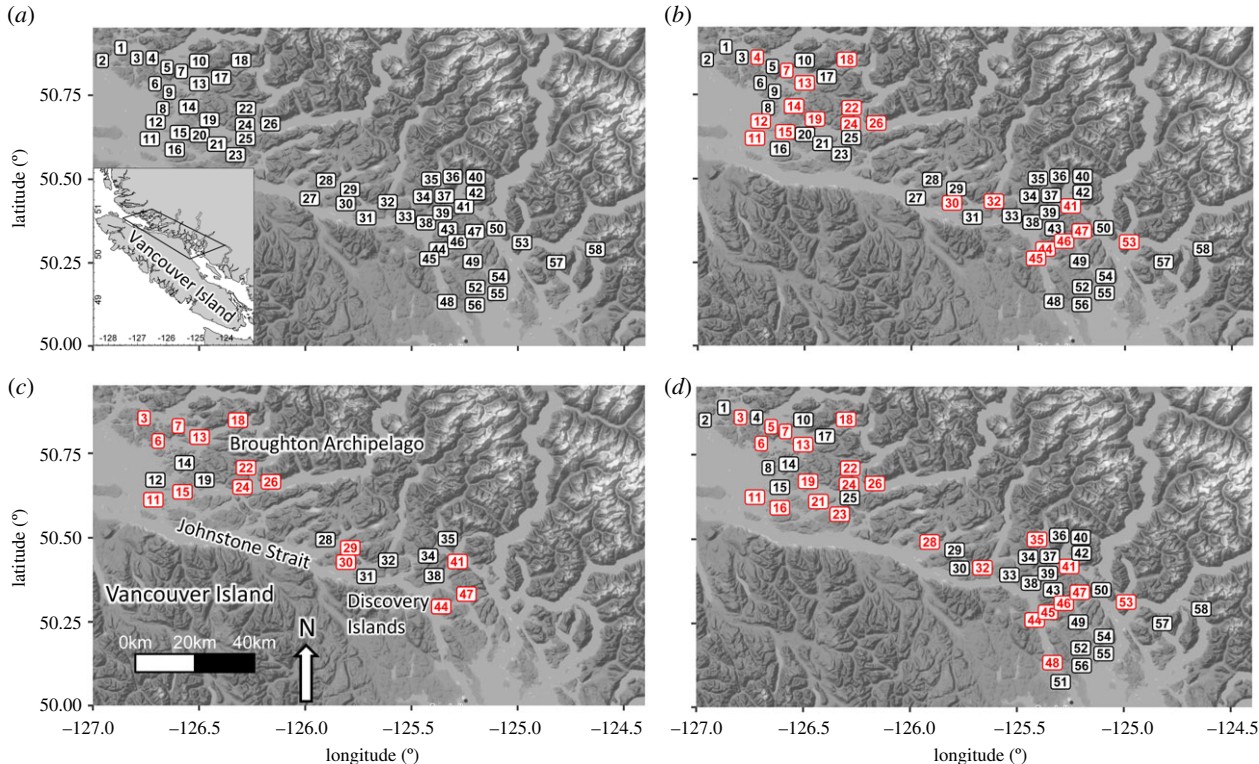

**Figure 1.** A map depicting (*a*) all farm tenure sites included in this study as well as individual maps for each sampling year, depicting (*b*) 2016, (*c*) 2017, and (*d*) 2018 sampling sites. Label colours in *b*, *c*, and *d* indicate which sites contained active salmon farms (red) and those that were inactive (black) at the time of sampling. Numbers correspond to site numbers in electronic supplementary material, table S1, which contains information on site status for each sampling year. (Online version in colour.)

biodiversity conservation, the extent to which reservoir hosts alter parasite communities has been assessed empirically in very few systems, natural or domestic, and only for a limited range of parasites.

Tracking the dynamics of multi-host parasite transmission has been a key challenge for marine aquaculture where domesticated fish may function as a reservoir host for parasites in coastal environments [8,9]. In response to seafood demand and plateaued fisheries [7,9,10], salmon aquaculture has expanded rapidly, and the abundance of domesticated salmon now exceeds wild populations in all global production regions where they coexist and global production of domestic salmon exceeds wild catch [10,11]. Transmission of macroparasites between wild and domestic populations has been well studied in salmonid aquaculture. For instance, transmission of native ectoparasitic copepods, sea lice (*Lepiopheirus salmonis* and *Caligus* spp.), from domesticated salmon can elevate infection rates of wild juvenile salmon [12,13] and impair the fitness of individuals [14,15] as well as overall recruitment [16–18]. Despite an extensive body of literature investigating pathogens (bacterial, viral, eukaryotic) on salmon farms and in wild fish populations [19–21], there are few studies exploring these pathogens in the environment where they can be shed and acquired by domestic and wild fish populations (but see [22,23]). Pathogen transmission from domesticated salmon is a concern for the conservation of wild salmon in Canada, including coho (*Oncorhynchus kisutch*), Chinook (*Oncorhynchus tshawytscha*), and sockeye (*Oncorhynchus nerka*) as well as wild populations of Atlantic salmon and sea trout in other regions where salmonid aquaculture encroaches on wild habitat [24–28]. The extent of transmission of this diverse assemblage of pathogens shared by wild and farmed salmon remains largely unknown.

In this paper, we use eDNA to empirically evaluate the distribution of a diverse assemblage of pathogens in the marine environment in relation to domestic salmon populations in coastal British Columbia (BC), Canada. In this system, transmission among salmonids can occur between five species of wild Pacific salmon and net-pen farmed Atlantic salmon (*Salmo salar*) via their shared marine environment. Salmonids can also contract pathogen infections from wild non-salmonid hosts. Our study area contains 58 salmon farm tenures, each of which can be stocked with over one million domesticated Atlantic salmon that spend approximately 18 months in marine net pens before being harvested (figure 1). In BC, farms culture a single year-class of salmon and are generally fallowed one to three months before being re-stocked with a new year-class of Atlantic salmon. The salmon farms in this region are situated on the marine migration routes of wild Pacific salmon from southern BC and Washington State, USA, including the Fraser River which supports Canada's largest populations of wild Pacific salmon. Over 3 years (2016–2018), we analysed the occurrence of 39 pathogens, including viral, bacterial, and eukaryotic agents, at subsets of these 58 salmon farm locations, a fraction of which are fallowed at any given time.

## 2. Methods

### (a) Sample collection

We collected samples of seawater at active and inactive salmon farm locations between Vancouver Island and the BC mainland over three years (2016–2018; figure 1). Inactive sites included both aquaculture sites fallowed between regular production cycles and sites which had been zoned for salmon farming but

have not been active for greater than three years. We conducted surveys in 2016 (March–April), 2017 (July–August), and 2018 (June–July) where we collected seawater from salmon-farm tenure locations (both active and inactive) in southern BC (electronic supplementary material, table S1). Inactive tenures were those locations approved for salmon aquaculture, but which did not contain farmed salmon at the time of sampling and served as reference sites. Approximately half of the sites were inactive each year, and many sites varied in their active versus inactive status among years (electronic supplementary material, table S1). We sampled 57, 24, and 53 sites of which 20, 15, and 24 were active, in 2016, 2017, and 2018, respectively (electronic supplementary material, table S1).

## (b) Sample processing

To isolate the community of cellular microbes and viruses in seawater collections, 10–12 l samples were filtered in the field immediately after collection and the viral fraction was stabilized via addition of an iron flocculant. Subsequently, in the laboratory, nucleic acids were extracted and DNA (or complementary DNA (cDNA) synthesized from RNA viruses) was analysed via quantitative polymerase chain reaction (PCR) targeting 39 species of pathogens as well as eDNA of Atlantic salmon (see electronic supplementary methods for details of seawater collection and filtration protocols; electronic supplementary material, figure S1 and table S2; S3 for a schematic of methods, methodological variation between sampling years, and assay details for surveyed pathogens).

## (c) Statistical analysis

For our statistical analyses, we included data from only those pathogen species that were encountered at least once in a survey (14 in 2016, 15 in 2017, and 19 in 2018). We modelled the presence of pathogen DNA as a binary variable (present or absent) per site using a generalized linear mixed-effects model (GLMM) with a binomial error distribution, that included fixed effects for farm activity (active versus empty) or Atlantic salmon eDNA (continuous), and random effects for site (intercept) and species (intercept and slope). The model was fitted for all sampling years combined by adding a fixed effect of sampling year on the intercept. To quantify the Atlantic salmon eDNA covariate, we calculated the inverse of Atlantic salmon eDNA cycle threshold (Ct) values before centring values on zero and dividing by their standard deviation. To evaluate whether small variations in water volumes influenced the results, the volumes processed were also included as a covariate for both pathogen eDNA detection, and Atlantic salmon eDNA levels. We fitted the GLMMs using the 'glmmTMB' package in R [29,30] and we evaluated support for each model using Akaike's information criterion [31].

We quantified the strength and direction of the association between pathogen eDNA detections and salmon farm status (active versus inactive) and relative Atlantic salmon eDNA concentration based on the odds ratios derived from the GLMM model fits. In both cases, an odds ratio equal to one suggests no association; whereas, an odds ratio less than one indicates a negative association between the predictor and pathogen detections and greater than one indicates a positive association. For binary predictors (e.g. salmon farm status), odds ratios represent the proportional change in the likelihood of detecting a pathogen associated with one level of the predictor relative to the alternative level of the predictor. For continuous predictor variables (e.g. Atlantic salmon eDNA), odds ratios represent the proportional change in the likelihood of detecting a pathogen per standard-deviation increase in Atlantic salmon eDNA concentration.

# 3. Results

## (a) Pathogen eDNA

Over the three years of surveys, we detected eDNA from 22 of the 39 pathogen species screened in our samples (figure 2; electronic supplementary material, table S4). In 2016, 2017, and 2018, we detected 14 of 37, 15 of 28, and 18 of 28 species of pathogens, respectively. Among the species that we detected, few were common, and the majority of species were detected in fewer than 20% of tested samples; three pathogen species were represented by a single detection in a single sampling year (figure 2; electronic supplementary material, table S4). It should be noted that one of the species represented by only a single detection (*Ichthyophonus hoferi*) was only assessed in 2016. The majority of virus positive detections were observed in the cell-associated (greater than 0.22 µm) sample fraction; however, three viruses including cutthroat trout virus (CTv-2), Atlantic salmon Calicivirus (ASCV), and viral encephalopathy and retinopathy (VER) exhibited the opposite pattern; whereby, they were more commonly detected in the sub-cellular (less than 0.22 µm) sample fraction (electronic supplementary material, table S5).

Across all years, we found that both farm activity and Atlantic salmon eDNA were positively correlated with pathogen eDNA detections. Models of farm activity and Atlantic salmon eDNA estimated the probability of detecting a pathogen was 2.72 (95% CI: 1.48, 5.02) times higher at active salmon farms relative to inactive sites and 1.76 (95% CI: 1.28, 2.42) times higher per standard deviation increase in Atlantic salmon eDNA variation, respectively (figure 3; electronic supplementary material, tables S6; S7). We found no association between pathogen eDNA detections and the volume of seawater filtered, and there was only small variation in the filtration volumes (see electronic supplementary material, tables S8; S9). Additionally, there were no significant differences in water temperature, salinity, or water clarity between active and inactive sites within a season (electronic supplementary material, table S8).

Across all three years of sampling, the same four agents: *Candidatus Syngamydia* salmonis, *Desmozoon lepeohtherii* (syn: *Paranucleospora theridion*), *Piscirickettsia salmonis*, and *Erythrocytic necrosis virus*, were the most commonly detected (figure 2; electronic supplementary material, table S4). By adding the random effect estimates for each pathogen species to the fixed effect estimate from each model, we estimated species-level associations between farm status or Atlantic eDNA and the probability of detecting pathogen eDNA. We found that *Moritella viscosa* and *Tenacibaculum maritimum* exhibited the largest positive correlation with domestic salmon populations based on both site status and Atlantic salmon eDNA models (figure 3). For most pathogen species, the estimated effects of site status and Atlantic salmon eDNA were in the same direction (figure 3).

# 4. Discussion

Reservoir host populations can increase the risk of disease transmission to, and extinction (or extirpation) risk of, nearby host populations [1,3]. Our study provides a comprehensive multi-species empirical analysis of pathogen eDNA in the nearby environment in relation to domesticated salmon populations. These data indicate that for many of

Proc. R. Soc. B 287: 20202010

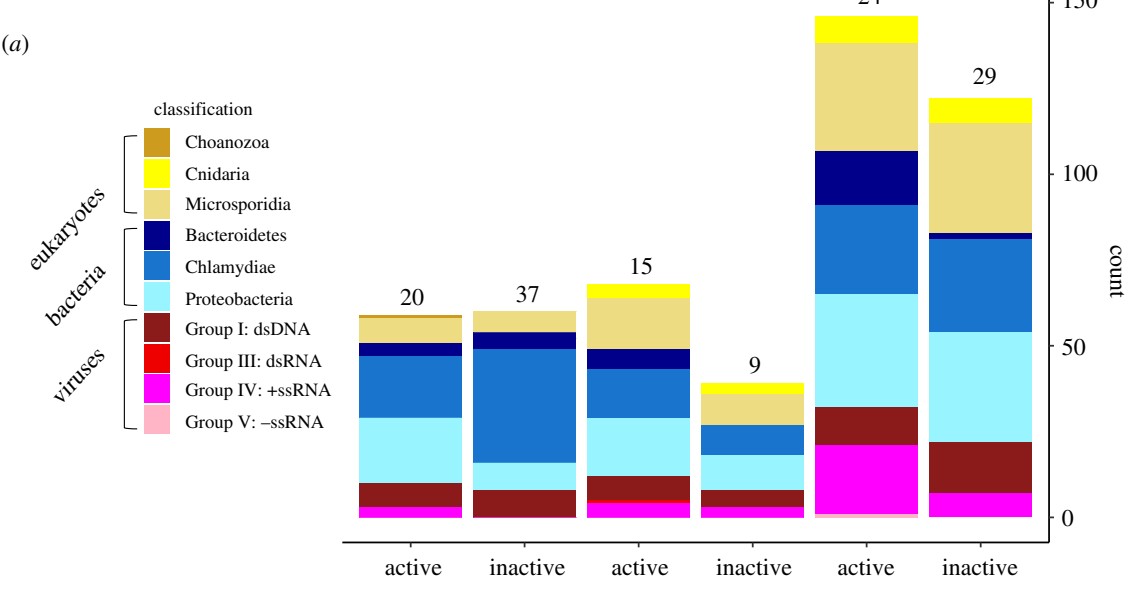

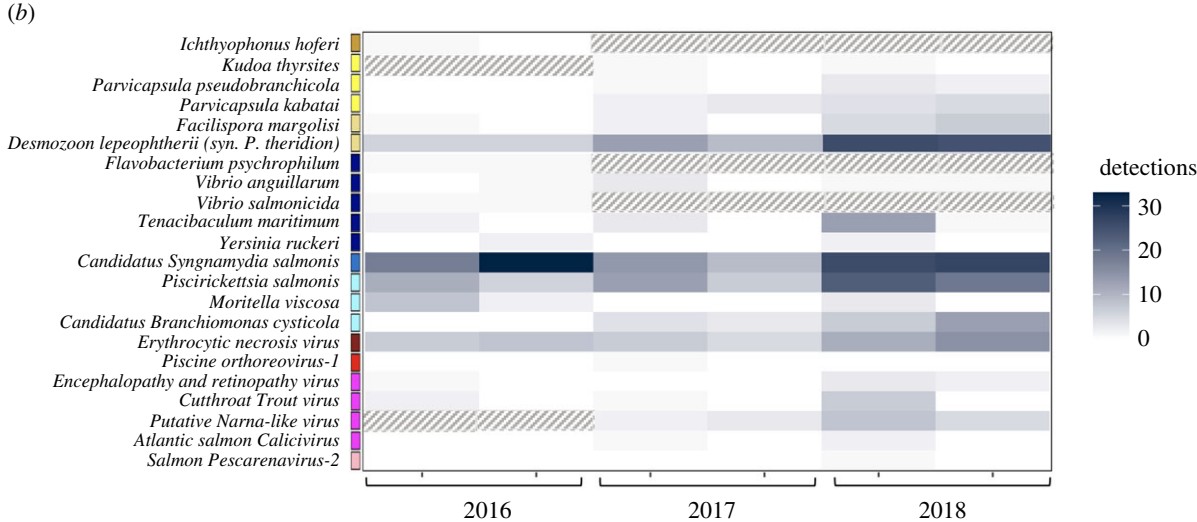

**Figure 2.** Raw qPCR detections of pathogen eDNA separated by phylum (*a*) and subdivided by species (*b*) at active and inactive sites across all three sampling years (2016–2018). Coloured boxes next to pathogen species indicate the phylum (eukaryotes and bacteria) or Baltimore class (viruses) classification group that each pathogen belongs to. The colour of pathogen species' cells (*b*) indicates the number of detections of each species by site status and sampling year. Cells with a pattern fill indicate pathogen species that were not tested in that sampling year. Numbers above stacked bars (*a*) indicate the number of active and inactive sites sampled in each year. (Online version in colour.)

the species assessed in our study, pathogen eDNA was positively correlated with domesticated salmon populations. Previous studies have empirically validated the relationship between gene copies and viable organisms for some species of fish and pathogens [32,33]; however, this study relied solely on eDNA and, therefore, we cannot explicitly relate the occurrence of pathogens to infection risk. Nonetheless, based on studies of eDNA degradation in marine coastal environments, which estimate eDNA half-life at between 6.9 and 72 h [34], the eDNA distribution we observed are likely correlated with the distribution of viable pathogens. It follows then, that salmon farms likely elevate the pathogen exposure of nearby wild fish populations via their shared environment.

Farmed salmon have previously been shown to function as a reservoir host for ectoparasitic sea lice, elevating infection risk in the marine environment over a radius of approximately 15 km per farm [13]. Our results indicate that salmon farms are associated with elevated pathogen

eDNA, leading us to propose that they may similarly act as a source of elevated pathogens for a diverse assemblage of pathogen species. However, the extent of dispersal and the duration of viability for each pathogen species remains to be characterized. In the marine environment, pathogen survival is highly variable ranging from the rapid decay of some RNA viruses to the weeks-long viability of some bacteria outside their host [35]. Also, temperature and UV radiation are known to affect destruction rates of pathogens in a species-specific manner [36–38]. For example, interactions with environmental microbiota and exposure to UV radiation can accelerate environmental degradation of infectious hematopoietic necrosis virus (IHNV), reducing the density of viable viruses by 2–6 orders of magnitude over a 3 h period [37]. Presumably, such factors could influence our ability to detect different pathogens in environmental samples.

We found that the probability of detecting pathogens was on average 2.72 times greater in the environment near active salmon farms when compared with control (inactive) sites,

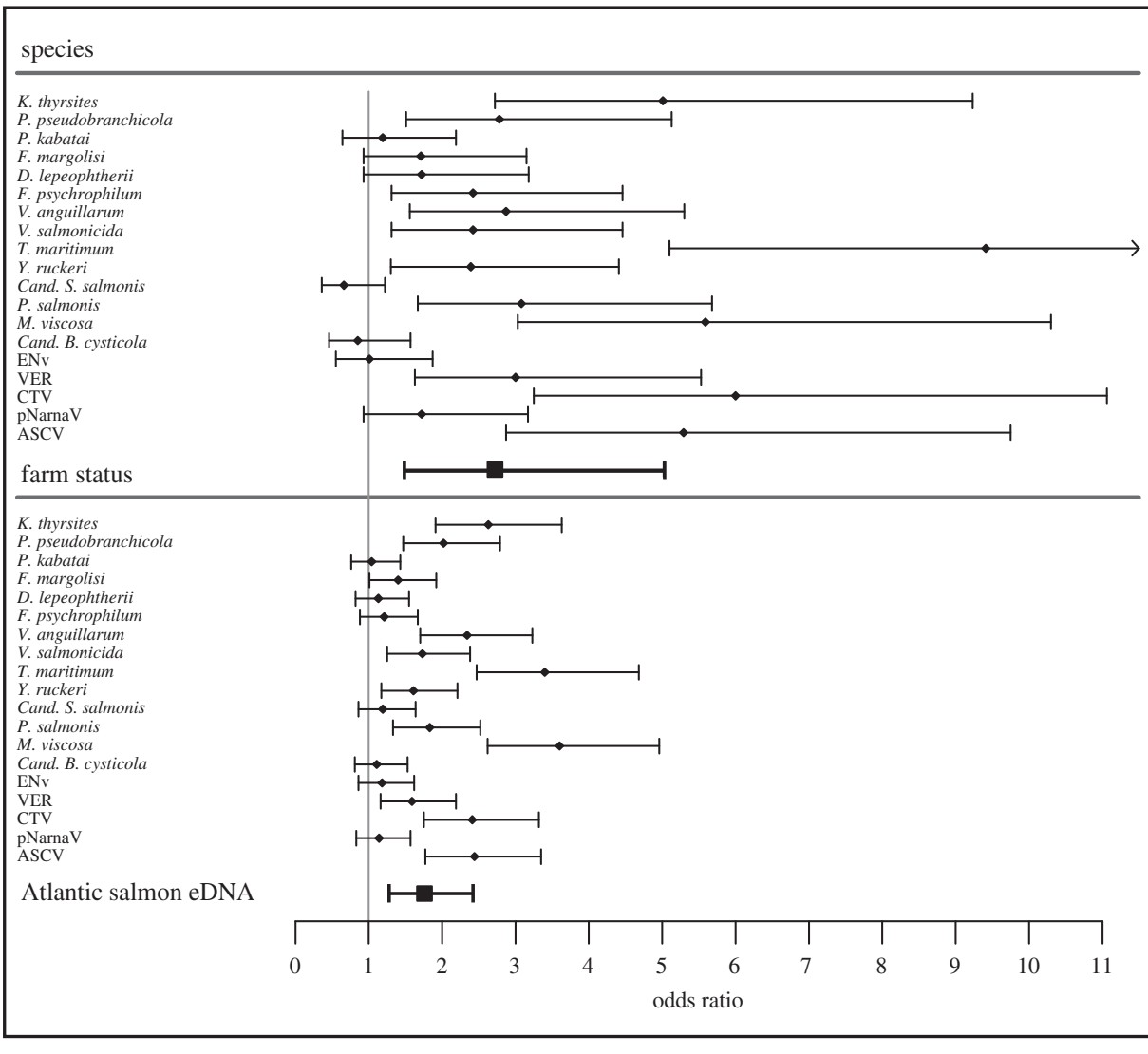

**Figure 3.** Pathogen eDNA detection odds ratios in relation to active salmon farms and relative Atlantic salmon eDNA concentration. Pathogen-specific odds ratios were generated by adding species-specific random effect estimates to the fixed effect predictor (i.e. farm status or Atlantic salmon eDNA) from multi-year GLMMs. Error bars depict pathogen-specific odds ratios $\pm$ fixed effect (farm status or Atlantic salmon eDNA) confidence intervals. Odds ratios for farm status and Atlantic salmon eDNA were generated from multi-year model estimates. Species estimates could not be estimated for pathogens that were represented by a single eDNA detection (*I. hoferi*, *PRV*, *SPAV-2*). Odds ratios from Atlantic salmon eDNA model results depict the change in probability of pathogen eDNA detection associated with a single standard deviation change in Atlantic salmon eDNA concentration.

with consistent estimation accuracy among years but lower precision in 2017 when the sample size was smaller (electronic supplementary material, table S7). These results also indicate that the detection of pathogens at a site increased with the relative concentration of Atlantic salmon eDNA measured at that site (electronic supplementary material, table S7). If pathogen eDNA is a reliable indicator of the presence of viable pathogens, these results would suggest there is an increased infection risk for wild salmon at locations of active salmon farms. However, pathogens vary in their infectivity and progression to cause disease in wild salmon depending on species, host condition, and environmental factors; therefore, further studies will be required to quantify the potential impacts of this environmental pathogen exposure on wild populations.

We found that Atlantic salmon eDNA was a better predictor of pathogen presence than farm status (figure 3; electronic supplementary material, table S6). Unsurprisingly, because Atlantic salmon do not naturally occur in coastal BC waters, the relative concentration of Atlantic salmon eDNA

was closely associated with salmon farm activity. However, the concentration of Atlantic salmon eDNA would also be more integrative of pathogen production and dispersal between connected sites that together combine to affect local infection risk. The dispersal of Atlantic salmon eDNA from active to inactive sites is supported by our data that detected Atlantic salmon eDNA at unstocked sites (2016: sites 39, 48; 2017: sites 19, 28, 31, 35; 2018: sites 4, 14, 15, 17, 25, 29, 30, 33, 34, 39, 43, 49, 50, 51, 52, 54, 56, 57, 58) in 25 instances with an average cycle threshold (Ct)-difference of 6.1, indicating approximately 100-fold differences in eDNA concentrations between active and inactive sites. Microscopic pathogens, like Atlantic salmon cells, can disperse throughout the environment, away from their source. Nonetheless, Atlantic salmon eDNA was observed more frequently and at higher concentrations near active salmon farms, suggesting that the spatial distribution of Atlantic salmon eDNA was closely tied to the distribution of active salmon farms. We expect pathogen eDNA to follow a similar pattern, whereby pathogen detections will occur most

frequently near their host source. Moreover, Atlantic salmon eDNA as a measure would also account for the differences between recently stocked farms (small fish equates to lower biomass and reduced Atlantic salmon eDNA) that would likely contain fewer pathogens than farms with maturing fish (see [21]).

We only assessed Atlantic salmon eDNA extracted from 0.22 μm filters, restricting our analysis to DNA associated with shed fish cells, which exhibit rapid decay relative to most microbes [38,39]. Variation among sites in the release, decay, and dispersal of host biological material, is governed by many of the same biotic and abiotic characteristics (e.g. UV, temperature, salinity, current, host metabolism) that influence the dissemination of pathogens from a point source [34,40,41]. As such, the distribution of Atlantic salmon eDNA concentrations is likely a good proxy for the speed and direction that passively dispersed pathogens are transported between sampling sites.

We normalized Atlantic salmon eDNA measurements within each sampling year to characterize spatial variation in eDNA, without confounding estimates with interannual variation in measurement precision. Accordingly, our estimates from Atlantic salmon eDNA models represent relative, not absolute, measures of the correlation between Atlantic salmon eDNA and pathogen eDNA. Due to the various biotic and abiotic factors that collectively affect measured concentrations of Atlantic salmon eDNA, further replication over space and time would be necessary to confidently estimate the absolute relationship between Atlantic salmon gene copies and pathogen gene copies. By normalizing Atlantic salmon eDNA measurements within each sampling season, we controlled for variation among sampling years in seasonal influences on the release and decay of Atlantic salmon eDNA. Therefore, we do not expect that our model estimates should be confounded by interannual variation in measured eDNA concentrations. However, this somewhat complicates the quantitative interpretation of eDNA model estimates, as these odds ratios reflect pathogen exposure in relation to variation in relative eDNA concentration, not absolute eDNA concentration.

Our results indicate that the increased frequency of pathogen detections at active sites is closely tied to variation in the abundance of Atlantic salmon, but perhaps also to aggregations of other fish species that can occur at active salmon farms [42]. Documentation of wild bycatch from net pens during the harvest of farmed salmon is common, and suggests that BC salmon farms may similarly attract some wild fish species, potentially altering disease transmission dynamics [43]. Pacific herring (Clupea pallasii) are one of the most commonly reported species of bycatch on salmon farms, and are competent hosts for a number of pathogens that infect farmed and wild salmon [44,45]. It is, therefore, possible that aggregation of herring or other wild fish at active sites may also contribute to elevated infection risk at those locations. This could help to explain the observed discrepancy between site status and Atlantic salmon eDNA as predictors of some pathogens (e.g. D. lepeophtherii), which appear to exhibit relatively strong associations with active salmon farms that are not supported by Atlantic salmon eDNA model estimates (figure 3). Based on previous observations of wild fish aggregating around aquaculture sites, we speculate that salmon farms may play an indirect role in facilitating the transmission of some infectious agents

among wild fish assemblages by attracting local aggregations of fish seeking food or shelter, a phenomenon which has been documented in other coastal aquaculture operations [42,46].

Wild fish species also likely contribute to the distribution of pathogen eDNA in the environment because many miocroparasites are generalists. For example, Erythrocytic necrosis virus (ENv) and Piscirickettsia salmonis were detected at a large number of sampling sites, in the presence and absence of salmon farms, suggesting that both domestic and wild hosts likely contributed to the distributions of eDNA we observed during this study (figure 2; electronic supplementary material, table S4). ENv is known to infect a large number of fish species, including Atlantic and Pacific salmon and Pacific herring [21,45,47–49]. Similarly, P. salmonis has been isolated from diverse cultured marine fish including both Atlantic and Pacific salmon, and Pacific herring (Miller K. 2019, unpublished data. (doi:10.1017/CBO9781107415324.004); [50]).

Our results also point to the role of abiotic seasonal variation as an important factor influencing the occurrence of pathogen eDNA. For example, Moritella viscosa is capable of infecting diverse fishes at temperatures below 8°C [51] and is implicated in winter ulcer disease in farmed salmonids [52]. Both the frequency and concentration of M. viscosa eDNA were higher in 2016 spring surveys (figure 2), when average water temperatures were 3°C lower than the average temperatures during the 2017 and 2018 summer surveys (electronic supplementary material, table S8). By contrast, D. lepeophtherii has a multi-host life cycle that includes sea lice [53,54] and requires warm water temperatures (greater than 10°C) to complete its development [55]. During our study, D. lepeophtherii eDNA was detected far less frequently in March–April 2016 surveys compared to surveys in July–August of 2017 and 2018 (figure 2). During sampling in 2016, the water was approximately 9°C and there are few natural adult salmon hosts present for sea lice [55], whereas in July–August of 2017 and 2018 when water temperatures were approximately 12°C and wild adult salmon are abundant. We speculate that this combination of biotic and abiotic factors may have contributed to the near ubiquitous distribution of D. lepeophtherii eDNA in 2017 and 2018, despite its relative rarity in 2016 (figure 2; electronic supplementary material, table S4).

For most pathogen species, we observed relative agreement in the magnitude and direction of model estimates of their eDNA association with site status and Atlantic salmon eDNA (figure 3). However, in two cases (Cand. S. salmonis, Cand. B. cysticola), the sign of estimates differed between the activity status of a farm and Atlantic salmon eDNA (figure 3). This apparent discrepancy could have arisen by chance, particularly for pathogens with low detection frequencies. Some less common eukaryotic (Ichthyophonus hoferi, Kudoa thyrsites, Facilispora margolisi), bacterial (Flavobacterium psychrophilum, Yersinia ruckerii, Vibrio anguillarum, Vibrio salmonicida), and viral pathogens (encephalopathy and retinopathy virus (ERv), Atlantic salmon calicivirus, salmon pescarenavirus (SPAv), Piscine orthoreovirus (PRv-1)) were detected at three or fewer sites in a given season (figure 2; electronic supplementary material, table S4). The source of these species is less obvious, although all have been observed in farmed salmonids [21,28].

In our analyses, the models treated salmon farms as active (containing fish), or empty; which ignores the history of the site and age of the fish. We speculate that this may have

decreased observable differences between active and inactive sites, as some inactive sites were recently fallowed and active sites with recently stocked young farmed fish may not have yet accumulated some microparasitic infections. Many pathogens are capable of persisting in the environment and on farm equipment for extended periods in the absence of their hosts [38], and the presence of wild reservoir hosts could delay the clearance of these species at recently fallowed sites. Another uncertainty is the potential contribution of genetic material from non-viable pathogens.

Our results relied on qPCR detection of gene fragments to explore variation in the abundances of hosts and pathogens. It is likely that a fraction of the genetic material that we detected in seawater samples was associated with non-viable pathogens; although, we suspect that such variation should be spatially correlated with the abundance of viable organisms for a given pathogen species. The environmental stability of pathogens, the efficacy of nucleic acid recovery and purification, and the relative instability of RNA compared to DNA could all contribute to detection biases among pathogens. For example, we collected samples near the surface, which might reduce the chance of detecting viruses that are more sensitive to UV degradation. We focused our analysis of the directly filtered cellular (greater than 0.22 µm) fraction on DNA for bacterial and eukaryotic agents, and our analysis of the chemically flocculated 'free virus' (less than 0.22 µm) sample fraction on viruses (RNA and DNA). In both cases, we did not enzymatically remove RNA or DNA. Hence, for the free virus fraction, cDNA was synthesized from a portion of the RNA extract, and the remainder was used to assess DNA viruses. For 2017 and 2018 samples, we included cell-associated RNA viruses in our analysis by synthesizing cDNA from both the cellular (greater than 0.22 µm) and subcellular (less than 0.22 µm) sample fractions, and found that most viral detections were present in the cell-associated fraction (electronic supplementary material, table S5). Without explicitly evaluating these and other potential sources of pathogen detection bias, we cannot conclude that species which we failed to observe were not present. This uncertainty is inherent in many eDNA studies when eDNA distributions cannot be evaluated against *a priori* expectations, given the suite of factors, both natural and methodological, which can influence the distribution and subsequent recovery of genetic material in environmental samples [56,57]. Although the sensitivity of our methodology is likely species specific, we do not expect that variations in the sensitivity of our species-specific assays were confounded with farm status.

One pathogen, PRv-1, has been reported at high prevalence in farmed Atlantic and Pacific salmon populations (average prevalence less than 70%; [21]), with generally low, but variable prevalence in wild salmon [21,27,58,59]. Interestingly, this agent was detected only once in our data, and only in the cellular fraction. In addition to technical issues for lack of detection, there are potentially other biological explanations for lack of detection. First, PRv is a virus that prefers cold water, and in Norway causes disease over the winter period [60]. Hence, we would expect that viral replication and shedding might be strongest in cooler water. Moreover, there is some evidence supporting the supposition that transmission potential of this virus varies over time [61,62]. Second, Norwegian studies have shown that the primary transmission route for the virus is faecal/oral [63] with

viral shedding through the removal of damaged infected erythrocytes [59,64]. As faecal material sinks at a rate of 4–6 cm sec$^{-1}$, it is also possible that faecal matter was not well represented in samples at 2 m depth [65]. However, as juvenile salmon incidentally consume faeces [63], their contact rate may be higher than is measured by our surface-oriented eDNA samples. Given these sources of variation, the low prevalence of PRv, and potentially other RNA or faecal transmitted viruses from our data may not indicate transmission potential. Future studies will need to examine this more carefully.

Despite the fact that we did not have paired pathogen prevalence data in farmed fish for comparison, we can compare our results to a previous assessment [21] of pathogen prevalence on salmon farms in this region. Although such comparisons cannot inform which pathogens would be expected to be present at specific sites, we can evaluate the relative agreement between host prevalence and environmental pathogen data on the epidemiology of each of the detected pathogens. For example, Laurin *et al*. [21] identified *D. lepeophtherii* as the most prevalent agent in farmed Atlantic salmon, with 88% of fish testing positive for this agent during their 2011–2013 surveys BC surveys. This is consistent with our assessment of *D. lepeophtherii*. This species was the second most frequently detected pathogen in our surveys (detected at 63% of sampled sites) and the estimated species effect from the multi-year model indicated a positive correlation with active salmon farms (figure 3). Additionally, Laurin *et al*. [21] reported that erythrocytic necrosis virus was among the most commonly detected viruses, but was more common on Chinook (37%) than Atlantic salmon (13%) farms. Erythrocytic necrosis virus was the most commonly observed virus in our surveys (40% of sites) but was not correlated with active salmon farms across years (figures 2 and 3). The bacterial pathogen *Cand. B. cysticola* was the most commonly detected bacterial pathogens on Chinook salmon farms (89%) reported in [21], but prevalence on Atlantic salmon farms was considerably lower (10%); this bacterium was commonly detected in our seawater collections but was not significantly correlated with active salmon farms or Atlantic salmon eDNA (figure 3).

Many of the agents that were detected at low frequency (less than 10%) in Atlantic salmon farms in [21] were not detected in our study. These include *Nucleospora salmonis*, *Loma salmonae*, and *Paramoeba perurans*. One freshwater bacterium [66], *Flavobacterium psychrophillum*, only occasionally observed in [21] was also observed twice in our samples. We suspect this agent may occur from stocking juvenile Atlantic salmon from hatcheries. Alternately, it may be associated with areas with more freshwater influence or when there is migration of wild juvenile salmon into the marine environment. The bacterial pathogen, *Renibacterium salmoninarum*, was present in 25% of farmed Atlantic salmon in [21] but not detected in any of our samples. Like PRv, this bacterium is transmitted horizontally via the faecal/oral route [67]; due to a rapid sinking rate, it is possible our surface-oriented samples may not be optimal for detection of faecal cellular matter. There are other explanations for its absence as well, such as improvements to monitoring, treatment, and site fallowing protocols in broodstock and marine production phases [66].

The shedding of pathogens from infected hosts depends upon the host tissue in which pathogens localize and often

also on the progression of infection, both of which may have contributed to the differences between pathogen prevalence from [21] and those observed in seawater samples during our environmental surveys. Furthermore, some pathogens exhibit distinct tissue tropisms at different stages of infection, and in different hosts, often resulting in variable infectiousness among infected hosts [58,59,64]. Ultimately, the parallels we draw between this study and others [21] are speculative due to the complex relationship between host–parasite biology and environmental transmission as well as the lack of overlap in time between the study of Laurin *et al.* [21] (collected 2011–2013) and this study (2016–2018). Further exploration of the relationship between host (domestic and wild) and environmental pathogen prevalence in this environment may enable more effective management.

The prevalence of infection and disease depends not only on the presence of the parasite but also on characteristics of the host individuals, community, and the environment. A number of laboratory studies have explored the relationship between host infection and pathogen release, as well as how pathogen exposure translates to host infection [63,68]. However, environmental variation is also important in mediating infection risk in natural systems, particularly in aquatic systems. For example, factors such as temperature, salinity, and the microbial community can influence pathogen decay and host susceptibility [37,69,70]. Additionally, simultaneous exposure to multiple stressors or pathogens can modulate host defences and alter infection kinetics and transmission dynamics [71].

Our study adds to a growing body of empirical work testing predictions from community epidemiology to better understand domestic-wildlife interactions [1,2,72,73]. Our results showing the presence of pathogen eDNA around active and fallowed farms suggests that domestic and wild reservoir host populations both contribute to pathogen signatures in a shared environment. If eDNA results are indicative of infectious pathogen communities, the results of this study underscore the complex nature of disease transmission in a coastal marine ecosystem. This research also potentially sheds light on the roles played by domestic and wild host reservoir populations in the distribution of pathogens in the marine environment of coastal BC. Our findings provide a first step in understanding how salmon farms may influence infection risk for declining and imperilled wild salmon populations such as BC's Fraser River sockeye, Chinook, and coho salmon [24,28,74]. Overall, our results suggest that encroachment of domestic populations on or within habitats of wild populations may create new interactions among wild and domestic populations via diverse, shared infectious agents. The ability to identify host populations that are at risk of infection from reservoir host populations, and the ecological processes that mitigate this risk, are important considerations for conservation management.

Data accessibility. The data and code used to conduct the analyses reported in this manuscript are available through the Dryad Digital Repository https://doi.org/10.5061/dryad.r7sqv9s98 [75].

Authors' contributions. D.S., A.B., L.O., S.S., and M.K. contributed to study conception and design. D.S., A.B., J.V., N.F., and M.K. contributed to sample collection. D.S., S.L., A.T., A.S., K.M., and S.S. contributed to molecular analysis. D.S., A.B., G.M., B.C., S.S., and M.K. contributed to statistical analysis and generation of figures. D.S. wrote the manuscript, D.S., A.B., G.M., L.O., J.V., N.F., B.C., K.M., S.S., and M.K. contributed equally to manuscript revisions.

Competing interests. We declare we have no competing interests.

Funding. This work was supported by funding from the David Suzuki Foundation and Ontario Graduate Scholarship to D.S., NSERC and Killam Trusts to A.B., the Liber Ero Fellowship Program to G.M., an NSERC Discovery grant to S.S., an NSERC Discovery grant, and Canada Research Chair to M.K.

Acknowledgements. We are grateful for fieldwork help from The David Suzuki Foundation, Alexandra Morton, Jodi Campbell, Amber Stroeder, Kiran Wadhawan, Emma Atkinson, Clare Atkinson, and Luke Rogers in addition to the staff and volunteers of the Salmon Coast Field Station, and Michael Staniewski, Andrew Long, and Karia Kaukinen who helped in the laboratory. This work received 2 years of fieldwork assistance from the Sea Shepherd Conservation Society and their research vessel the Martin Sheen, and we are grateful for the help and support of the crew, in particular, Marc Archambault and Francois Martin for captaining the vessel, and Carolina Castro and Eva Hidalgo for coordinating sampling trips.

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
