## [Reviewer comments · Proceedings of the Royal Society B: Biological Sciences]

Review History

RSPB-2020-1327.R0 (Original submission)

Review form: Reviewer 1

Recommendation

Reject – article is scientifically unsound

Scientific importance: Is the manuscript an original and important contribution to its field?

Marginal

General interest: Is the paper of sufficient general interest?

Good

Quality of the paper: Is the overall quality of the paper suitable?

Marginal

Is the length of the paper justified?

Yes

Should the paper be seen by a specialist statistical reviewer?

Yes

Do you have any concerns about statistical analyses in this paper? If so, please specify them explicitly in your report.

No

It is a condition of publication that authors make their supporting data, code and materials available - either as supplementary material or hosted in an external repository. Please rate, if applicable, the supporting data on the following criteria.

Is it accessible?

Yes

Is it clear?

Yes

Is it adequate?

Yes

Do you have any ethical concerns with this paper?

No

Comments to the Author

Manuscript # RSPB-2020-1327, "Enrichment of infectious disease agents in the marine environment associated with salmon farms", is a well-designed and executed comparison of host and pathogen eDNA in two types of environments: active vs fallowed fish farms in British Columbia. It is apparent that a lot of work went into this project over a 3-year experimental period, when the authors assessed the waterborne presence of eDNA from Atlantic salmon and 39 microparasite spp. from 58 salmon farm sites. If the paper would have stuck to the well-supported conclusions that host and pathogen eDNA differed between active and fallowed farm sites, the paper would be on solid footing. However, serious over-extensions of their findings occur throughout the manuscript and are bolstered by misinterpretations or conflation of standard terminology in disease ecology. I'm afraid that the primary findings of the paper (involving reservoirs and elevated risk of transmission near farms) are simply not supported by an objective interpretation of the results. These statements may be correct, but they were neither tested nor confirmed with this experimental design. For example, a simple correlation between pathogen eDNA, host eDNA, and farm status is not sufficient for assigning reservoir status to either Atlantic salmon or the farms; the salmon were never even sampled in this study.

Specific comments:

Line 1: The title is not an accurate reflection of the work that was performed. It is unclear, what is meant by the word "Enrichment", but its inclusion here is confusing. The word means "the act of making fuller, more meaningful, or rewarding." Also, the authors did not assess "infectious agents"; rather they assessed eDNA. There is a very serious difference between these two terms; they are not synonymous. Throughout the manuscript, the term "pathogen eDNA" is repeatedly conflated with terms like "pathogen", "parasite", "virus", "microparasite species", "microparasite detections", "pathogen exposure", "microparasite exposure", and "infectious agent". It must be made clear that pathogen eDNA was detected.

A more objective title would be: "Differences in host and pathogen eDNA between active and fallowed salmon farms"

The word "fallowed" rather than "inactive" is recommended throughout the manuscript. Use of the word "inactive" implies that the farm is decommissioned. This is not the case in this experimental system. Rather, the farms are simply fallowed for a period until they are re-activated.

Line 63: Replace "dwarfs" with "exceeds"

Lines 63 – 74: This stated difference between knowledge of macro and micro-parasites in aquaculture is somewhat contrived and misleading. Micro-parasites have been studied as well,

or perhaps better than macroparasites in aquaculture. Granted, the authors have worked on sea lice macroparasites, but please don't dismiss the decades of work done by others on microparasites. A quick lit search would refute this position. Please eliminate this section.

Line 78: It is unclear what "open" net pen farmed Atlantic salmon are. Perhaps just eliminate the word "open."

Lines 89-92: Remove this sentence; do not state your conclusions in the Introduction section.

Line 96: If the water was collected 20M away from the net pen (Suppl. Line 10), then it does not really represent a sample from an active pen. Why weren't the samples collected from inside the pens?

Figure 1: As displayed, it is very difficult to cross reference between Figure 1 and Suppl Table 1 to determine the temporal and spatial proximity between active and fallow net pens. It would be nice to know that active and fallowed sites were sampled from the same geographic and temporal proximity.

Figure 2: This is the essence of the paper. After removing the conflating language of pathogen, parasite, etc., from the paper and appropriately discussing this figure in terms of fish pathogen eDNA, one must question the novelty of detecting higher levels of eDNA from fish and fish pathogens in focal areas where fish are concentrated. Wouldn't one expect to also see the same pattern in focal areas containing aggregations of wild fish (eg, eDNA from sardines and sardine pathogens in large schools of free ranging sardines)?

Line 239: Again, none of the criteria for a reservoir host were satisfied in this study:

- 1) Live pathogens were not detected, only eDNA.
- 2) Neither pathogens nor pathogen DNA were detected in Atlantic salmon, the proposed reservoir host.
- 3) Transmission was neither examined nor demonstrated in this study.

Line 241: This study does not show "loading of the marine environment by domesticated salmon. This is an over-reach.

Lines 243-245: "These data indicate that many of the microparasite species assessed in this study may be maintained by domesticated salmon and elevated in the marine environment through which salmon migrate." This statement should be removed; this study simply did not test or evaluate this hypothesis.

Lines 247-251: Again, this is an overreach of the results and should be removed.

Line 270-271: How does the detection of eDNA translate into "increased infection risk for wild salmon at locations of active farms"? Again, this is an overreach.

Lines 284-286: In the same manner that the presence of Atlantic salmon eDNA (as was detected in some of the fallowed pens) does not demonstrate the presence of live Atlantic salmon, the presence of pathogen eDNA does not demonstrate the presence of live pathogens. You must be very careful to not over-extend the interpretation of your results.

Lines 386-389: The primary route of transmission for most of the viruses that were screened for ENv, Piscine Parvo, Piscine Orthoreo, etc), all are transferrer through the water, not oral - fecal. Also, it is curious why some of the most common viurses of the region and host (VHSV, IHNV, and IPNV) were not assessed (Suppl Table 3)?

Line 401: This statement is somewhat misleading. You are correct, one can not prove a negative; however, you also cannot not demonstrate that eDNA detections equate to detections of live pathogens. Seems as though evidence is being selected to support a particular story line and omitted when it refutes the story.

Line 421: Replace "actively" with "incidentally."

Lines 455 - 456: BKD transmission is not exclusively oral - fecal. This is a relatively minor transmission route. It seems as though the literature is being cherry picked to support pre-conceived hypotheses.

Lines 458-460: How do you explain the detection of Kudoa eDNA which should be located well-inside the tissues of the fish and not necessarily shed?

Review form: Reviewer 2

Recommendation

Accept with minor revision (please list in comments)

Scientific importance: Is the manuscript an original and important contribution to its field?

Excellent

General interest: Is the paper of sufficient general interest?

Excellent

Quality of the paper: Is the overall quality of the paper suitable?

Excellent

Is the length of the paper justified?

Yes

Should the paper be seen by a specialist statistical reviewer?

No

Do you have any concerns about statistical analyses in this paper? If so, please specify them explicitly in your report.

No

It is a condition of publication that authors make their supporting data, code and materials available - either as supplementary material or hosted in an external repository. Please rate, if applicable, the supporting data on the following criteria.

Is it accessible?

Yes

Is it clear?

Yes

Is it adequate?

Yes

Do you have any ethical concerns with this paper?

No

Comments to the Author

Shea et al. report the results of an impressive qPCR study of microparasites on farmed Atlantic salmon in British Columbia, across three years and 58 sites. Importantly, those sites were either active salmon farms or fallow ones, providing a control for the natural experiment. The team brings to bear qPCR primers for 39 different pathogens, and on the whole, finds that active salmon farms tend to be reservoirs for certain of these diseases. The study is interesting and careful, and I hope to see it in press soon.

I have just a few comments on the piece.

- Replication. The methods suggest 3x subsampling to create replicates (or pseudo-replicates, if you prefer), and there were likely qPCR technical replicates as well, but I don't see any of those data in the (outstandingly simple and clean) supplementary data or discussed in the text.

- Signal:Noise. I am guessing, and a quick look at the supplemental data bears this out, that the data are noisy, insofar as apparent trends change year-to-year and species-to-species. Hence, the

strongest results (as given in the abstract, for example) are overall changes in detection probability, lumping all species and years. Which makes sense, but then, the results section disaggregates the logistic models into individual years, which makes it kind of difficult to interpret the overall gist of things. Figure 3, also, seems to give the take-home message, and it represents aggregated data. I wonder if the authors would consider plotting the logistic models (combined and year-to-year, or hierarchical with year as a random effect), to illustrate the lack of consistency among years.

- Figure 2 should be corrected for effort. Of course there are more total parasites detected in 37 samples of inactive sites in 2016 than in 20 samples of active sites — they represent totals of nearly twice the sampling effort.

- Re: figure 3, this may be my lack of substantial experience in reading odds-ratio plots, but I find it hard to interpret the odds ratio of “farm status” and “Atlantic Salmon eDNA”. Moreover, re: Atlantic Salmon eDNA in that same figure, are the data plotted the change in odds ratio when salmon eDNA increases by one standard deviation?

- Lines 404-425, it might make sense for the authors to do some occupancy modeling here to figure out the rates of true and false detection for each assay + species. That would be easy, given the data structure (they could treat sites as replicates of a common phenomenon) and would do away with the need to speculate here about these parameters.

- Occupancy modeling would then inform lines 434-457 in a useful way.

- Model selection: I think, supplementary table 6, the authors are comparing apples and oranges (although I could be wrong here). The year-specific models have only (roughly) one-third of the data each, and so their AIC values are necessarily lower than the combined-year model. I don't think this is a fair comparison, although I admit, I'm not sure I know how to do such a comparison in a fair way. My guess is that the all-years model is a better idea, but the authors are certainly more familiar with their data than I am.

Decision letter (RSPB-2020-1327.R0)

01-Jul-2020

Dear Mr Shea:

I am writing to inform you that your manuscript RSPB-2020-1327 entitled "Enrichment of infectious disease agents in the marine environment associated with salmon farms" has, in its current form, been rejected for publication in Proceedings B.

This action has been taken on the advice of referees, who have recommended that substantial revisions are necessary. With this in mind we would be happy to consider a resubmission, provided the comments of the referees are fully addressed. However please note that this is not a provisional acceptance.

1) A 'response to referees' document including details of how you have responded to the comments, and the adjustments you have made.

- 2) A clean copy of the manuscript and one with 'tracked changes' indicating your 'response to referees' comments document.
- 3) Line numbers in your main document.

Sincerely,
 Dr Daniel Costa
 mailto: proceedingsb@royalsociety.org

Associate Editor
 Board Member: 1

Comments to Author:

Thank you for allowing us to review this interesting MS. While both reviewers agree that the study is well-designed, Reviewer 1 highlights a major weakness of the paper in its current iteration: it extrapolates well beyond the bounds of the data collected, drawing conclusions about putative disease reservoirs and elevated risk of transmission near salmon farms that cannot be inferred from the eDNA data collected. If the authors are willing to substantially roll back their far-reaching inferences and to make the other changes suggested by the two reviewers, we would be happy to review a new version of the manuscript.

Reviewer(s)' Comments to Author:
 Referee: 1

Comments to the Author(s)

Manuscript # RSPB-2020-1327, "Enrichment of infectious disease agents in the marine environment associated with salmon farms", is a well-designed and executed comparison of host and pathogen eDNA in two types of environments: active vs fallowed fish farms in British Columbia. It is apparent that a lot of work went into this project over a 3-year experimental period, when the authors assessed the waterborne presence of eDNA from Atlantic salmon and 39 microparasite spp. from 58 salmon farm sites. If the paper would have stuck to the well-supported conclusions that host and pathogen eDNA differed between active and fallowed farm sites, the paper would be on solid footing. However, serious over-extensions of their findings occur throughout the manuscript and are bolstered by misinterpretations or conflation of standard terminology in disease ecology. I'm afraid that the primary findings of the paper (involving reservoirs and elevated risk of transmission near farms) are simply not supported by an objective interpretation of the results. These statements may be correct, but they were neither tested nor confirmed with this experimental design. For example, a simple correlation between pathogen eDNA, host eDNA, and farm status is not sufficient for assigning reservoir status to either Atlantic salmon or the farms; the salmon were never even sampled in this study.

Specific comments:

Line 1: The title is not an accurate reflection of the work that was performed. It is unclear, what is meant by the word "Enrichment", but its inclusion here is confusing. The word means "the act of making fuller, more meaningful, or rewarding." Also, the authors did not assess "infectious agents"; rather they assessed eDNA. There is a very serious difference between these two terms; they are not synonymous. Throughout the manuscript, the term "pathogen eDNA" is repeatedly conflated with terms like "pathogen", "parasite", "virus", "microparasite species", "microparasite detections", "pathogen exposure", "microparasite exposure", and "infectious agent". It must be made clear that pathogen eDNA was detected.
 A more objective title would be: "Differences in host and pathogen eDNA between active and fallowed salmon farms"

The word “fallowed” rather than “inactive” is recommended throughout the manuscript. Use of the word “inactive” implies that the farm is decommissioned. This is not the case in this experimental system. Rather, the farms are simply fallowed for a period until they are re-activated.

Line 63: Replace “dwarfs” with “exceeds”

Lines 63 – 74: This stated difference between knowledge of macro and micro-parasites in aquaculture is somewhat contrived and misleading. Micro-parasites have been studied as well, or perhaps better than macroparasites in aquaculture. Granted, the authors have worked on sea lice macroparasites, but please don’t dismiss the decades of work done by others on microparasites. A quick lit search would refute this position. Please eliminate this section.

Line 78: It is unclear what “open” net pen farmed Atlantic salmon are. Perhaps just eliminate the word “open.”

Lines 89-92: Remove this sentence; do not state your conclusions in the Introduction section.

Line 96: If the water was collected 20M away from the net pen (Suppl. Line 10), then it does not really represent a sample from an active pen. Why weren’t the samples collected from inside the pens?

Figure 1: As displayed, it is very difficult to cross reference between Figure 1 and Suppl Table 1 to determine the temporal and spatial proximity between active and fallow net pens. It would be nice to know that active and fallowed sites were sampled from the same geographic and temporal proximity.

Figure 2: This is the essence of the paper. After removing the conflating language of pathogen, parasite, etc., from the paper and appropriately discussing this figure in terms of fish pathogen eDNA, one must question the novelty of detecting higher levels of eDNA from fish and fish pathogens in focal areas where fish are concentrated. Wouldn’t one expect to also see the same pattern in focal areas containing aggregations of wild fish (eg, eDNA from sardines and sardine pathogens in large schools of free ranging sardines)?

Line 239: Again, none of the criteria for a reservoir host were satisfied in this study:

- 1) Live pathogens were not detected, only eDNA.
- 2) Neither pathogens nor pathogen DNA were detected in Atlantic salmon, the proposed reservoir host.
- 3) Transmission was neither examined nor demonstrated in this study.

Line 241: This study does not show “loading of the marine environment by domesticated salmon. This is an over-reach.

Lines 243-245: “These data indicate that many of the microparasite species assessed in this study may be maintained by domesticated salmon and elevated in the marine environment through which salmon migrate.” This statement should be removed; this study simply did not test or evaluate this hypothesis.

Lines 247-251: Again, this is an overreach of the results and should be removed.

Line 270-271: How does the detection of eDNA translate into “increased infection risk for wild salmon at locations of active farms”? Again, this is an overreach.

Lines 284-286: In the same manner that the presence of Atlantic salmon eDNA (as was detected in some of the fallowed pens) does not demonstrate the presence of live Atlantic salmon, the presence of pathogen eDNA does not demonstrate the presence of live pathogens. You must be very careful to not over-extend the interpretation of your results.

Lines 386-389: The primary route of transmission for most of the viruses that were screened for (ENv, Piscine Parvo, Piscine Orthoreo, etc), all are transferrable through the water, not oral – fecal. Also, it is curious why some of the most common viruses of the region and host (VHSV, IHNV, and IPNV) were not assessed (Suppl Table 3)?

Line 401: This statement is somewhat misleading. You are correct, one can not prove a negative; however, you also cannot not demonstrate that eDNA detections equate to detections of live pathogens. Seems as though evidence is being selected to support a particular story line and omitted when it refutes the story.

Line 421: Replace “actively” with “incidentally.”

Lines 455 – 456: BKD transmission is not exclusively oral – fecal. This is a relatively minor transmission route. It seems as though the literature is being cherry picked to support pre-conceived hypotheses.

Lines 458-460: How do you explain the detection of *Kudoa* eDNA which should be located well-inside the tissues of the fish and not necessarily shed?

Referee: 2

Comments to the Author(s)

Shea et al. report the results of an impressive qPCR study of microparasites on farmed Atlantic salmon in British Columbia, across three years and 58 sites. Importantly, those sites were either active salmon farms or fallow ones, providing a control for the natural experiment. The team brings to bear qPCR primers for 39 different pathogens, and on the whole, finds that active salmon farms tend to be reservoirs for certain of these diseases. The study is interesting and careful, and I hope to see it in press soon.

I have just a few comments on the piece.

- Replication. The methods suggest 3x subsampling to create replicates (or pseudo-replicates, if you prefer), and there were likely qPCR technical replicates as well, but I don't see any of those data in the (outstandingly simple and clean) supplementary data or discussed in the text.
- Signal:Noise. I am guessing, and a quick look at the supplemental data bears this out, that the data are noisy, insofar as apparent trends change year-to-year and species-to-species. Hence, the strongest results (as given in the abstract, for example) are overall changes in detection probability, lumping all species and years. Which makes sense, but then, the results section disaggregates the logistic models into individual years, which makes it kind of difficult to interpret the overall gist of things. Figure 3, also, seems to give the take-home message, and it represents aggregated data. I wonder if the authors would consider plotting the logistic models (combined and year-to-year, or hierarchical with year as a random effect), to illustrate the lack of consistency among years.
- Figure 2 should be corrected for effort. Of course there are more total parasites detected in 37 samples of inactive sites in 2016 than in 20 samples of active sites — they represent totals of nearly twice the sampling effort.
- Re: figure 3, this may be my lack of substantial experience in reading odds-ratio plots, but I find it hard to interpret the odds ratio of “farm status” and “Atlantic Salmon eDNA”. Moreover, re: Atlantic Salmon eDNA in that same figure, are the data plotted the change in odds ratio when salmon eDNA increases by one standard deviation?
- Lines 404-425, it might make sense for the authors to do some occupancy modeling here to figure out the rates of true and false detection for each assay + species. That would be easy, given the data structure (they could treat sites as replicates of a common phenomenon) and would do away with the need to speculate here about these parameters.
- Occupancy modeling would then inform lines 434-457 in a useful way.
- Model selection: I think, supplementary table 6, the authors are comparing apples and oranges (although I could be wrong here). The year-specific models have only (roughly) one-third of the data each, and so their AIC values are necessarily lower than the combined-year model. I don't think this is a fair comparison, although I admit, I'm not sure I know how to do such a comparison in a fair way. My guess is that the all-years model is a better idea, but the authors are certainly more familiar with their data than I am.

Author's Response to Decision Letter for (RSPB-2020-1327.R0)

See Appendix A.

RSPB-2020-2010.R0

Review form: Reviewer 1

Recommendation

Accept as is

Scientific importance: Is the manuscript an original and important contribution to its field?

Acceptable

General interest: Is the paper of sufficient general interest?

Acceptable

Quality of the paper: Is the overall quality of the paper suitable?

Good

Is the length of the paper justified?

Yes

Should the paper be seen by a specialist statistical reviewer?

No

Do you have any concerns about statistical analyses in this paper? If so, please specify them explicitly in your report.

No

It is a condition of publication that authors make their supporting data, code and materials available - either as supplementary material or hosted in an external repository. Please rate, if applicable, the supporting data on the following criteria.

Is it accessible?

Yes

Is it clear?

Yes

Is it adequate?

Yes

Do you have any ethical concerns with this paper?

No

Comments to the Author

The authors have adequately addressed my initial comments and questions. Only minor stylistic and editorial comments remain; these can be addressed by the typesetter.

Decision letter (RSPB-2020-2010.R0)

14-Sep-2020

Dear Mr Shea

I am pleased to inform you that your manuscript RSPB-2020-2010 entitled "Environmental DNA (eDNA) from multiple pathogens is elevated near active Atlantic salmon farms" has been accepted for publication in Proceedings B.

The referee(s) have recommended publication, but also suggest some minor revisions to your manuscript. Therefore, I invite you to respond to the referee(s)' comments and revise your manuscript. Because the schedule for publication is very tight, it is a condition of publication that you submit the revised version of your manuscript within 7 days. If you do not think you will be able to meet this date please let us know.

In order to ensure effective and robust dissemination and appropriate credit to authors the dataset(s) used should be fully cited. To ensure archived data are available to readers, authors

should include a 'data accessibility' section immediately after the acknowledgements section. This should list the database and accession number for all data from the article that has been made publicly available, for instance:

Sincerely,
Dr Daniel Costa
<mailto:proceedingsb@royalsociety.org>

Associate Editor
Comments to Author:

Thank you for doing such a thorough job of incorporating the suggestions from the reviewers. The manuscript is now substantially improved, and it is my pleasure to accept it for publication in Proc B.

Reviewer(s)' Comments to Author:

Referee: 1

Comments to the Author(s).

The authors have adequately addressed my initial comments and questions. Only minor stylistic and editorial comments remain; these can be addressed by the typesetter.

Decision letter (RSPB-2020-2010.R1)

22-Sep-2020

Dear Mr Shea

I am pleased to inform you that your manuscript entitled "Environmental DNA (eDNA) from multiple pathogens is elevated near active Atlantic salmon farms" has been accepted for publication in Proceedings B.

Open Access

Paper charges

Sincerely,

Appendix A

Associate Editor

Board Member: 1

Comments to Author:

Thank you for allowing us to review this interesting MS. While both reviewers agree that the study is well-designed, Reviewer 1 highlights a major weakness of the paper in its current iteration: it extrapolates well beyond the bounds of the data collected, drawing conclusions about putative disease reservoirs and elevated risk of transmission near salmon farms that cannot be inferred from the eDNA data collected. If the authors are willing to substantially roll back their far-reaching inferences and to make the other changes suggested by the two reviewers, we would be happy to review a new version of the manuscript.

Author Response:

We are grateful for the opportunity to submit a revised version of our paper and thank you for the fast turnaround on the first round of reviews. We found the reviewer comments to be helpful and valuable. We have considered them carefully and have revised the manuscript accordingly. In particular, we have been more careful with our use of terminology (e.g. pathogen vs pathogen eDNA) and we are now clearer about what conclusions are directly supported by the data versus the conceptual context within which our conclusions are situated. We respond to the reviewer comments below on a point by point basis.

Reviewer(s)' Comments to Author:

Referee: 1

Comments to the Author(s)

Manuscript # RSPB-2020-1327, "Enrichment of infectious disease agents in the marine environment associated with salmon farms", is a well-designed and executed comparison of host and pathogen eDNA in two types of environments: active vs fallowed fish farms in British Columbia. It is apparent that a lot of work went into this project over a 3-year experimental period, when the authors assessed the waterborne presence of eDNA from Atlantic salmon and 39 microparasite spp. from 58 salmon farm sites. If the paper would have stuck to the well-supported conclusions that host and pathogen eDNA differed between active and fallowed farm sites, the paper would be on solid footing. However, serious over-extensions of their findings occur throughout the manuscript and are bolstered by misinterpretations or confluations of standard terminology in disease ecology. I'm afraid that the primary findings of the paper (involving reservoirs and elevated risk of transmission near farms) are simply not supported by an objective interpretation of the results. These statements may be correct, but they were neither tested nor confirmed with this experimental design. For example, a simple correlation between pathogen eDNA, host eDNA, and farm status is not sufficient for assigning reservoir status to either Atlantic salmon or the farms; the salmon were never even sampled in this study.

Author Response:

Thank you for the detailed and helpful review of our manuscript. Upon reading your comments and reflecting on our first version of the manuscript, we recognize the need to be more careful with our wording, and in particular to avoid conflating pathogen with pathogen eDNA. We also recognise the need to be clearer about what conclusions are directly supported by the data (differences in eDNA at active vs inactive sites) versus the conceptual context of reservoirs and transmission risk within which our conclusions are situated. We have carefully edited the manuscript accordingly, and respond to your specific comments point-by-point below.

Specific comments:

Reviewer 1 Comment 1:

Line 1: The title is not an accurate reflection of the work that was performed. It is unclear, what is meant by the word "Enrichment", but its inclusion here is confusing. The word means "the act of making fuller, more meaningful, or rewarding."

Response to Reviewer 1 Comment 1:

Although we used the word 'enrichment' in the sense of 'making fuller', we agree that this could be misread, and the title could be improved. We therefore changed the title to "Environmental DNA (eDNA) from multiple pathogens is elevated near active Atlantic salmon farms" to more accurately reflect the key findings of our study.

Reviewer 1 Comment 2:

Also, the authors did not assess "infectious agents"; rather they assessed eDNA. There is a very serious difference between these two terms; they are not synonymous. Throughout the manuscript, the term "pathogen eDNA" is repeatedly conflated with terms like "pathogen", "parasite", "virus", "microparasite species", "microparasite detections", "pathogen exposure", "microparasite exposure", and "infectious agent". It must be made clear that pathogen eDNA was detected.

Response to Reviewer 1 Comment 2:

We agree that detecting pathogen eDNA is not equivalent to detecting viable infectious pathogens. The conflation of terms was not intended to be misleading, as we thought our meaning was clear, if implied. While there is convention in the literature to refer to eDNA detections by only organism names (refs below), we appreciate the importance being explicit, and we have edited the manuscript accordingly. We now refer to "pathogen eDNA" rather than "pathogens" in all instances where this was needed. For example:

Lines 225-227: "Our study provides a comprehensive multi-species empirical analysis of pathogen eDNA in relation to domesticated salmon populations. These data indicate that for many of the species assessed in this study pathogen eDNA was positively correlated with domesticated salmon populations."

Lines 232-236: "Nonetheless, based on studies of eDNA degradation in marine coastal environments, which estimate eDNA half-life at between 6.9 and 72 hours [34], the eDNA distribution observed in this study is likely correlated with the distribution of viable pathogens. It follows then, that salmon farms likely elevate the pathogen exposure of nearby wild fish populations via their shared environment."

Lines 239-241: "Our results indicate that salmon farms are associated with elevated pathogen eDNA, leading us to propose that they may similarly act as a source of elevated pathogens for a diverse assemblage of species."

Lines 256-258: "If pathogen eDNA is a reliable indicator of the presence of viable pathogens, these results would suggest there is an increased infection risk for wild salmon near active salmon farms."

Here are a few examples of eDNA studies where the authors extrapolate from gene fragments to organisms when describing their results and conclusions:

1. Robinson CV, Uren Webster TM, Cable J, James J, Consuegra S. 2018 Simultaneous detection of invasive signal crayfish, endangered white-clawed crayfish and the crayfish plague pathogen using environmental DNA. *Biol. Conserv.* **222**, 241–252. This study described the distribution of native and non-native crayfish species as well as an invasive fungal pathogen using eDNA.
2. Sato Y, Mizuyama M, Sato M, Minamoto T, Kimura R, Toma C. 2019 Environmental DNA metabarcoding to detect pathogenic *Leptospira* and associated organisms in leptospirosis-endemic areas of Japan. *Sci. Rep.* **9**, 1–11. This study utilized an eDNA metabarcoding approach to identify the presence of pathogenic *Leptospira* bacteria in water samples.
3. Bastos Gomes G, Hutson KS, Domingos JA, Chung C, Hayward S, Miller TL, Jerry DR. 2017 Use of environmental DNA (eDNA) and water quality data to predict protozoan parasites outbreaks in fish farms. *Aquaculture* **479**, 467–473. (doi:10.1016/j.aquaculture.2017.06.021): This study used eDNA to quantify the abundance of protozoan *Chilodonella hexasticha* in freshwater barramundi farms.

4. Fossøy F, Brandsegg H, Sivertsgård R, Pettersen O, Sandercock BK, Solem Ø, Hindar K, Mo TA. 2020 Monitoring presence and abundance of two gyrodactylid ectoparasites and their salmonid hosts using environmental DNA. *Environ. DNA* 2, 53–62. (doi:10.1002/edn3.45): This study found that concentrations of trematode (*Gyrodactylus salaris*) eDNA were indicative of parasite abundance.
5. Nardi CF, Fernández DA, Vanella FA, Chalde T. 2019 The expansion of exotic Chinook salmon (*Oncorhynchus tshawytscha*) in the extreme south of Patagonia: an environmental DNA approach. *Biol. Invasions* 21, 1415–1425. (doi:10.1007/s10530-018-1908-8): This study confirmed the range expansion of exotic Chinook salmon in Patagonia on the basis of eDNA detection.

Reviewer 1 Comment 3:

A more objective title would be: “Differences in host and pathogen eDNA between active and fallowed salmon farms”

Response to Reviewer 1 Comment 3:

We changed the title to “Environmental DNA (eDNA) from multiple pathogens is elevated near active Atlantic salmon farms” to more accurately reflect the results and conclusions of this study.

Reviewer 1 Comment 4:

The word “fallowed” rather than “inactive” is recommended throughout the manuscript. Use of the word “inactive” implies that the farm is decommissioned. This is not the case in this experimental system. Rather, the farms are simply fallowed for a period until they are re-activated.

Response to Reviewer 1 Comment 4:

On this point we disagree. The inactive sites were a mixture of sites that were fallowed between regular production cycles as well as sites that were approved tenures for farming but were not active for longer periods of time for unknown (to us) reasons. We added a sentence to the methods to clarify this characterization of sites as either “inactive” or “active”.

Lines 97-99: “Inactive sites included both aquaculture sites fallowed between regular production cycles and sites which had been zoned for salmon farming but had not been active for greater than three years.”

Reviewer 1 Comment 5:

Line 63: Replace “dwarfs” with “exceeds”

Response to Reviewer 1 Comment 5:

We replaced “dwarfs” with “exceeds” (line 63)

Reviewer 1 Comment 6:

Lines 63 – 74: This stated difference between knowledge of macro and micro-parasites in aquaculture is somewhat contrived and misleading. Micro-parasites have been studied as well, or perhaps better than macroparasites in aquaculture. Granted, the authors have worked on sea lice macroparasites, but please don’t dismiss the decades of work done by others on microparasites. A quick lit search would refute this position. Please eliminate this section.

Response to Reviewer 1 Comment 6:

We agree and apologize. Our intention was not to downplay the extensive work that has been done on fish pathogens in both aquaculture and wild fish populations, but rather that there are fewer data on spread of pathogens from farmed to wild salmon populations than there are for macroparasites. We revised this sentence: (lines 69-72): “Despite an extensive body of literature investigating pathogens (bacterial, viral, eukaryotic) on salmon farms and in wild fish populations (e.g. [19-21]), there are few studies exploring the spread of these pathogens from farmed fish into the environment where they can be acquired by wild fish populations (but see

[22,23]).” to reflect the specific topic that this study seeks to address while highlighting a number of microparasite studies in wild and domestic fish populations.

Reviewer 1 Comment 7:

Line 78: It is unclear what “open” net pen farmed Atlantic salmon are. Perhaps just eliminate the word “open.”

Response to Reviewer 1 Comment 7:

We have removed the word open from “open net pen” throughout the manuscript.

Reviewer 1 Comment 8:

Lines 89-92: Remove this sentence; do not state your conclusions in the Introduction section.

Response to Reviewer 1 Comment 8:

We have removed reference to our results in the introduction section.

Reviewer 1 Comment 9:

Line 96: If the water was collected 20M away from the net pen (Suppl. Line 10), then it does not really represent a sample from an active pen. Why weren't the samples collected from inside the pens?

Response to Reviewer 1 Comment 9:

The objective of the study was to evaluate if salmon farms are associated with elevated pathogen eDNA in the marine environment nearby but outside of aquaculture pens, in waters that are potentially inhabited by wild fish. The resultant data are therefore relevant to how salmon farms alter infection risk for nearby wild fish, with the caveat that eDNA of pathogens does not necessarily equal viable infectious pathogens. We very much agree that it would be interesting to also have samples and data from seawater inside the pens and from the farmed salmon themselves, but that is well beyond the scope of this study.

Reviewer 1 Comment 10:

Figure 1: As displayed, it is very difficult to cross reference between Figure 1 and Suppl Table 1 to determine the temporal and spatial proximity between active and fallow net pens. It would be nice to know that active and fallowed sites were sampled from the same geographic and temporal proximity.

Response to Reviewer 1 Comment 10:

We agree, it is difficult to relate the geographic location of sites depicted in Figure 1 to sampling information (i.e. site status and sampling date) in Table S1. To address this, we have added to Figure 1 maps for each sampling year with site labels coloured according to their status (active vs. inactive) at the time of sampling. Additionally, we have reordered the numbering of sites in Table S1 based on the distance of each site from the Northwestern-most site. Now the order of sites in Table S1 approximately reflects their order of appearance on the map from top-left (Northwest) to bottom-right (Southeast). We have changed site numbers in Figure 1 and the attached supplementary data file to reflect this change. We think that these changes will make it easier to cross reference between the Figure 1 map and the Table S1 of site information. We have revised the manuscript where we specifically reference site numbers to reflect this change.

Lines 267-272: “The dispersal of Atlantic salmon eDNA from active to inactive sites is supported by our data that detected Atlantic salmon eDNA at unstocked sites (2016: sites 39, 48; 2017: sites 19, 28, 31, 35; 2018: sites 4, 14, 15, 17, 25, 29, 30, 33, 34, 39, 43, 49, 50, 51, 52, 54, 56, 57, 58) in twenty-five instances with an average Ct-difference of 6.1, indicating approximately 100-fold differences in eDNA concentrations between active and inactive sites.”

Reviewer 1 Comment 11:

Figure 2: This is the essence of the paper. After removing the conflating language of pathogen, parasite, etc., from the paper and appropriately discussing this figure in terms of fish pathogen eDNA, one must question the novelty of detecting higher levels of eDNA from fish and fish pathogens in focal areas where fish are concentrated.

Wouldn't one expect to also see the same pattern in focal areas containing aggregations of wild fish (e.g., eDNA from sardines and sardine pathogens in large schools of free ranging sardines)?

Response to Reviewer 1 Comment 11:

We completely agree. However, it remains a key unknown for management and policy if and to what extent pathogen species spread from farmed salmon into the marine environment where wild fish occur. This was emphasized, for example, in the recommendations of the Cohen Commission looking into the decline of wild Sockeye salmon from the Frazer River (Cohen, 2012). It is also likely of interest for many disease ecologists to understand which pathogens or pathogen groups can spread from domesticated to wild host populations, in order to better understand disease emergence and associated risk. Within that context, the primary goal of this work was to empirically evaluate whether salmon farms alter the local community of pathogens relative to inactive sites, using pathogen eDNA as a proxy for the pathogen community. Although the claim may appear intuitive, there are few data with which to empirically evaluate its extent and variation. The fact that eDNA from different pathogen species exhibited differential associations with active salmon farms suggests that there is biologically- and policy-relevant variation in spread and transmission risk among pathogen species. We hope that this work will serve as a foundation and that further studies can probe specific pathogen associations to determine which species' eDNA is most likely to be elevated on salmon farms as well as in areas where wild hosts are abundant and what host, parasite, environmental factors are most important in shaping these associations. Additionally, we hope that future studies will seek to reconcile the connection between pathogen eDNA and viable pathogens for these pathogen species in order to improve the interpretation of other similar eDNA-based surveys.

Reviewer 1 Comment 12:

Line 239: Again, none of the criteria for a reservoir host were satisfied in this study:

- 1) Live pathogens were not detected, only eDNA.
- 2) Neither pathogens nor pathogen DNA were detected in Atlantic salmon, the proposed reservoir host.
- 3) Transmission was neither examined nor demonstrated in this study.

Response to Reviewer 1 Comment 12:

We agree. We have carefully revised the text to clearly differentiate conclusions that are directly supported by the results from the broader conceptual context within which the results are situated. For example, the study does not demonstrate that farmed salmon are a reservoir host but rather tests the predicted spatial distribution of pathogens in the marine environment that would occur if farmed salmon served as a reservoir. As such, the pattern of pathogen eDNA observed in this study is consistent with salmon farms functioning as a reservoir for some of the pathogen species, but it does not directly demonstrate that they are a reservoir.

Line 232-236: Nonetheless, based on studies of eDNA degradation in marine coastal environments, which estimate eDNA half-life at between 6.9 and 72 hours [34], the eDNA distribution we observed are likely correlated with the distribution of viable pathogens. It follows then, that salmon farms likely elevate the pathogen exposure of nearby wild fish populations via their shared environment.

Reviewer 1 Comment 13:

Line 241: This study does not show "loading of the marine environment by domesticated salmon. This is an over-reach.

Response to Reviewer 1 Comment 13:

We have rephrased this sentence:

Lines 225-227: "Our study provides a comprehensive multi-species empirical analysis of pathogen eDNA in the nearby marine environment in relation to domesticated salmon populations."

Reviewer 1 Comment 14:

Lines 224-226: “These data indicate that many of the microparasite species assessed in this study may be maintained by domesticated salmon and elevated in the marine environment through which salmon migrate.” This statement should be removed; this study simply did not test or evaluate this hypothesis.

Response to Reviewer 1 Comment 14:

We have removed this statement.

Reviewer 1 Comment 15:

Lines 247-251: “The potential implications of such dynamics include apparent competition between wild populations of non-salmonid fish and wild salmon, elevated mortality and extirpation risk of small populations of wild salmon, and adaptation of diverse parasites to domesticated salmon populations leading to epidemics, or increased disease burden “

Again, this is an overreach of the results and should be removed.

Response to Reviewer 1 Comment 15:

We have removed this statement.

Reviewer 1 Comment 16: While these results translate into increased infection risk for wild salmon at locations of active salmon farms, microparasites vary in their infectivity and progression to cause disease in wild salmon depending on species, host condition, and environmental factors.

Line 270-271: How does the detection of eDNA translate into “increased infection risk for wild salmon at locations of active farms”? Again, this is an overreach.

Response to Reviewer 1 Comment 16:

We agree that our data on eDNA do not allow us to directly assess infection risk. We have revised these statements to reflect the conditions under which this inference may be valid.

Lines 256-258: “If pathogen eDNA is a reliable indicator of the presence of viable pathogens, these results would suggest there is an increased infection risk for wild salmon at locations of active salmon farms.”

Reviewer 1 Comment 17:

Lines 284-286: In the same manner that the presence of Atlantic salmon eDNA (as was detected in some of the fallowed pens) does not demonstrate the presence of live Atlantic salmon, the presence of pathogen eDNA does not demonstrate the presence of live pathogens. You must be very careful to not over-extend the interpretation of your results.

Response to Reviewer 1 Comment 17:

We agree. However, the intent of this sentence was to highlight the dispersal potential of biological material from active salmon farms. Similarly, viable pathogens and/or pathogen eDNA may travel long distances from their source via passive dispersal mediated by hydrodynamic processes.

Lines 272-277: “Microscopic pathogens, like Atlantic salmon cells, can disperse throughout the environment, away from their source. Nonetheless, Atlantic salmon eDNA was observed more frequently and at higher concentrations near active salmon farms, suggesting that the spatial distribution of Atlantic salmon eDNA was closely tied to the distribution of active salmon farms. We expect pathogen eDNA to follow a similar pattern, whereby pathogen eDNA detections should occur most frequently near their host source.”

Reviewer 1 Comment 18:

Lines 386-389: The primary route of transmission for most of the viruses that were screened for ENv, Piscine Parvo, Piscine Orthoreo, etc), all are transferred through the water, not oral – fecal. Also, it is curious why some of the most common viruses of the region and host (VHSV, IHNV, and IPNV) were not assessed (Suppl Table 3)?

Response to Reviewer 1 Comment 18:

We would have been interested in assessing VHSV, IHNV, and IPNV and their exclusion was not an oversight. These viruses are internationally reportable and are the prevue of the Canadian Food Inspection Agency (CFIA). Assays to these agents were approved for research purposes on the Fluidigm BioMark platform, but the agreement between DFO and the CFIA stipulated that they would only be employed by DFO staff on samples with DFO chain of custody. As this study was carried out by a graduate student working within the DFO molecular genetics laboratory, they did not meet these criteria.

In terms of prevalence of these viruses, DFO has compiled an extensive database on pathogen prevalence in aquaculture salmon. None of these viruses were detected in the 3,071 fish analysed in a longitudinal study across four farms, while IHNV was detected in 1/931 farm audit fish (farmed salmon are vaccinated against this virus) and VHSV was detected in 2/931 farm audit fish, so these viruses causing acute disease are not, in fact, commonly observed across the industry. IPNV has not been detected in any farmed fish in BC. Hence, we were confident that excluding these reportable agents would not significantly change our results, as we would have been unlikely to detect them in association with the farms. VHSV is, however, a common virus infecting herring.

Reviewer 1 Comment 19:

Line 401: This statement is somewhat misleading. You are correct, one cannot prove a negative; however, you also cannot not demonstrate that eDNA detections equate to detections of live pathogens. Seems as though evidence is being selected to support a particular story line and omitted when it refutes the story.

Response to Reviewer 1 Comment 19:

We have responded above re terminology of pathogen vs pathogen eDNA, which was an issue of terminology rather than an issue of false positives or false negatives of detecting eDNA. In this case of possible false negatives, we added an additional sentence to articulate this point more explicitly.

Lines 381-384: “This uncertainty is inherent in many eDNA studies when eDNA distributions cannot be evaluated against a priori expectations, given the suite of factors, both natural and methodological, which can influence the distribution and subsequent recovery of genetic material in environmental samples [57,58].”

Reviewer 1 Comment 20:

Line 421: Replace “actively” with “incidentally.”

Response to Reviewer 1 Comment 20:

We replaced “actively” with “incidentally” on line 398-400:

“However, as juvenile salmon incidentally consume feces [64], their contact rate may be higher than is measured by our surface-oriented eDNA samples.”

Reviewer 1 Comment 21:

Lines 455 – 456: BKD transmission is not exclusively oral – fecal. This is a relatively minor transmission route. It seems as though the literature is being cherry picked to support pre-conceived hypotheses.

Response to Reviewer 1 Comment 21:

This sentence just tries to explain the absence of one pathogen species (*Renibacterium salmonarium*) in our data – that pathogens shed in feces may sink below our sampling depths before they reach the sampling locations. There are other explanations for its absence as well, which we should have added, such as improvements in husbandry and treatment since its high prevalence in the 1990s. We have added these additional explanations to the revised manuscript. Lines 432-434: “There are other explanations for its absence as well, such as improvements to monitoring, treatment, and site following protocols in broodstock and marine production phases [68].”

Reviewer 1 Comment 22:

Lines 458-460: How do you explain the detection of Kudoa eDNA which should be located well-inside the tissues of the fish and not necessarily shed?

Response to Reviewer 1 Comment 22:

We agree that the shedding of *Kudoa thyrsites* from infected fish seems unlikely or at least likely to occur at low rates, given the tissue tropism of this parasite and that cohabitation challenge studies have been unsuccessful in generating horizontal transmission in Atlantic salmon. *Kudoa thyrsites* eDNA was detected only twice in our study. This is consistent with the expectation that *K. thyrsites* eDNA would rarely be detected in seawater. We would also note that *K. thyrsites* must exist somewhere in the environment to transmit at all. The most probable source of *K. thyrsites* in the marine environment is spores released from its unidentified marine intermediate host. However, *K. thyrsites* may also be released into the marine environment after myoliquifaction of farmed salmon carcasses.

Referee: 2

Comments to the Author(s)

Shea et al. report the results of an impressive qPCR study of microparasites on farmed Atlantic salmon in British Columbia, across three years and 58 sites. Importantly, those sites were either active salmon farms or fallow ones, providing a control for the natural experiment. The team brings to bear qPCR primers for 39 different pathogens, and on the whole, finds that active salmon farms tend to be reservoirs for certain of these diseases. The study is interesting and careful, and I hope to see it in press soon.

I have just a few comments on the piece.

Author response:

Thank you for the constructive feedback on our manuscript.

Reviewer 2 Comment 1:

- Replication. The methods suggest 3x subsampling to create replicates (or pseudo-replicates, if you prefer), and there were likely qPCR technical replicates as well, but I don't see any of those data in the (outstandingly simple and clean) supplementary data or discussed in the text.

Response to Reviewer 2 Comment 1:

All qPCR Ct values depicted in the supplementary data file represent the average from duplicate qPCR reactions. We only considered qPCR data for pathogens and Atlantic salmon eDNA if the target was positively detected in both qPCR reactions. We have now added columns to the revised supplementary data file containing raw Ct-values from each replicate qPCR reaction for pathogen species and Atlantic salmon eDNA.

Reviewer 2 Comment 2

- Signal:Noise. I am guessing, and a quick look at the supplemental data bears this out, that the data are noisy, insofar as apparent trends change year-to-year and species-to-species. Hence, the strongest results (as given in the abstract, for example) are overall changes in detection probability, lumping all species and years. Which makes sense, but then, the results section disaggregates the logistic models into individual years, which makes it kind of difficult to interpret the overall gist of things. Figure 3, also, seems to give the take-home message, and it represents aggregated data. I wonder if the authors would consider plotting the logistic models (combined and year-to-year, or hierarchical with year as a random effect), to illustrate the lack of consistency among years.

Response to Reviewer 2 Comment 2:

We have reflected on these comments and agree that the presentation of results from single year models distracts from the overall results and have revised the results to include only the multi-year model. We had previously included results from single-year models in Table S6 and Table S7 (but did not compare single-year models among years) to highlight that the same general patterns observed in aggregate, across sampling years, were also observed within each sampling year albeit with less statistical power. Although, the relative ranking of models (Table S6) differed between years, the estimates associated with Farm Status and Atlantic salmon eDNA were similar across single year models and multi-year models, but standard errors increased with reduced sample sites. The aggregate model accounts for sampling year; therefore, including the individual year models is redundant and distracting.

With respect to reviewer 2's comment regarding modeling sampling year as a random effect, this is something that we initially considered. Ultimately, we decided to model sampling year as a fixed effect, as we did not think that we had sufficient sampling year replication (three years) to robustly estimate the variance of the sampling year as a random effect. We have decided to therefore stay with the fixed-effect multi-year model in the paper, but it is worth noting here that the results of the year-random effects model is nonetheless very similar: for the random effects model we modeled site as a random effect nested within sampling year and obtained estimates for farm status (β :1.04; SE: 0.35) and Atlantic salmon eDNA models (β : 0.57; SE: 0.18) that were similar to farm status (β :1.00; SE: 0.31) and Atlantic salmon eDNA (β : 0.57; SE: 0.16) models where year was treated as a fixed effect. As such, there is no meaningful consequence to the results whether year is treated as a fixed versus random effect, and in our view, proceeding with the fixed effects model is statistically preferable due to low replication of year with which to estimate the variance of a year random effect.

Reviewer 2 Comment 3:

- Figure 2 should be corrected for effort. Of course, there are more total parasites detected in 37 samples of inactive sites in 2016 than in 20 samples of active sites — they represent totals of nearly twice the sampling effort.

Response to Reviewer 2 Comment 3:

We have revised the manuscript to be clearer as to the purpose of Figure 2 – it is a visual description of the distribution of the *raw data*, complete with information on sample effort (above each bar in the top panel). The figure is intended as a more engaging visual depiction of the raw qPCR data that are presented numerically in the Supplementary Table 4. Note that figure 3 depicts pathogen comparisons with respect to farm status and Atlantic salmon eDNA from a GLMM that does account for unequal sampling coverage. We have revised the figure caption for figure 2 to reflect the fact that it depicts raw qPCR results and not results from statistical comparisons. We appreciate reviewer 2 pointing out this ambiguity and hope that this revision will allow readers to more easily navigate the results.

Lines 185-186: "**Figure 2.** Raw qPCR detections of pathogen eDNA separated by phylum (A) and subdivided by species (B) at active and inactive sites across all three sampling years (2016-2018).

Reviewer 2 Comment 4:

- Re: figure 3, this may be my lack of substantial experience in reading odds-ratio plots, but I find it hard to interpret the odds ratio of "farm status" and "Atlantic Salmon eDNA". Moreover, re: Atlantic Salmon eDNA in that same figure, are the data plotted the change in odds ratio when salmon eDNA increases by one standard deviation?

Response to Reviewer 2 Comment 4:

We added an explanation of how to interpret odds ratios for binary and continuous predictors in the context of this study on lines (131-141): "We quantified the strength and direction of the association between pathogen eDNA detections and salmon farm status (active vs. inactive) and relative Atlantic salmon eDNA concentration based on the odds ratios derived from the GLMM model fits. In both cases, an odds ratio equal to one suggests no association; whereas, an odds ratio less than one indicates a negative association between the predictor and pathogen detections and greater than one indicates a positive association. For binary predictors (e.g. salmon farm status), odds ratios represent the proportional change in the likelihood of detecting a pathogen associated with one level of the predictor relative to the alternative level of the predictor. For continuous predictor variables (e.g. Atlantic salmon eDNA), odds ratios represent the proportional change in the likelihood of detecting a pathogen per standard-deviation increase in Atlantic salmon eDNA concentration."

We have also clarified in the figure 3 caption that Atlantic salmon eDNA odds ratios represent the proportional change in probability of detection associated with a one standard deviation increase in Atlantic salmon eDNA concentration.

Lines 214-216: “*Odds ratios from Atlantic salmon eDNA model results depict the change in probability of pathogen eDNA detection associated with a single standard deviation change in Atlantic salmon eDNA concentration.”

Reviewer 2 Comment 5:

- Lines 404-425, it might make sense for the authors to do some occupancy modeling here to figure out the rates of true and false detection for each assay + species. That would be easy, given the data structure (they could treat sites as replicates of a common phenomenon) and would do away with the need to speculate here about these parameters.
- Occupancy modeling would then inform lines 434-457 in a useful way.

Response to Reviewer 2 Comment 5:

We appreciate Reviewer 2's recommendation to utilize occupancy models to address the rates of false detections and false negatives for each pathogen species. We looked into the potential application of these models in the context of this study and it appears our data may not be a good fit for this modeling framework, as they violate some of the assumptions of occupancy models. For example, the time gap between our surveys was approximately one year. Occupancy models require that the time interval separating replicate sampling events is relatively short and that the true occupancy of a site does not change between replicate sampling events. Given the inherently episodic nature of infectious diseases, it is unlikely that the presence of a pathogen at any site would remain consistent across years. Additionally, there are twenty instances in our data where the salmon farm status (i.e. active salmon farm vs. inactive site) differed between replicate sampling events. Another assumption of occupancy models is that extinction/colonization rates are identical between sites or that heterogeneity can be adequately modeled. Given that the salmon farm tenures in this study reside in biodiverse and dynamic environments, the distribution of wild hosts contributing to the dissemination of pathogens changes over time and space which likely translates to differential exposure to pathogens of farm populations. We are therefore not confident that our data are suitable for occupancy modeling. Finally, the manuscript is already at the length limits of the journal and there is little scope to add new analysis within those limits.

Reviewer 2 Comment 6:

- Model selection: I think, supplementary table 6, the authors are comparing apples and oranges (although I could be wrong here). The year-specific models have only (roughly) one-third of the data each, and so their AIC values are necessarily lower than the combined-year model. I don't think this is a fair comparison, although I admit, I'm not sure I know how to do such a comparison in a fair way. My guess is that the all-years model is a better idea, but the authors are certainly more familiar with their data than I am.

Response to Reviewer 2 Comment 6:

This is now a moot point as we decided to take the year-specific models out in response to the reviewer's earlier comment. However, to explain, we included Table S6 to summarize all modeling results in a single table, but the model selection was conducted only within years for single-year models and among multi-year models for the multi-year models. So, the problem of comparing apples and oranges did not occur, but we understand how that could be misread as all the results were presented in the same table.

References

- Bastos Gomes G, Hutson KS, Domingos JA, Chung C, Hayward S, Miller TL, Jerry DR. 2017 Use of environmental DNA (eDNA) and water quality data to predict protozoan parasites outbreaks in fish farms. *Aquaculture* **479**, 467–473.
- Cohen, Honourable BI. 2012 *The Uncertain Future of Fraser River Sockeye Volume 1 • The Sockeye Fishery*.
- Collins RA, Wangensteen OS, O’Gorman EJ, Mariani S, Sims DW, Genner MJ. 2018 Persistence of environmental DNA in marine systems. *Commun. Biol.* **1**, 1–11.
- Cowart DA, Murphy KR, Cheng CHC. 2018 Metagenomic sequencing of environmental DNA reveals marine faunal assemblages from the West Antarctic Peninsula. *Mar. Genomics* **37**, 148–160.
- Fossøy F, Brandsegg H, Sivertsgård R, Pettersen O, Sandercock BK, Solem Ø, Hindar K, Mo TA. 2020 Monitoring presence and abundance of two gyrodactylid ectoparasites and their salmonid hosts using environmental DNA. *Environ. DNA* **2**, 53–62.
- Nardi CF, Fernández DA, Vanella FA, Chalde T. 2019 The expansion of exotic Chinook salmon (*Oncorhynchus tshawytscha*) in the extreme south of Patagonia: an environmental DNA approach. *Biol. Invasions* **21**, 1415–1425.
- Marshall WL, Sitjà-Bobadilla A, Brown HM, MacWilliam T, Richmond Z, Lamson H, Morrison DB, Afonso LOB. 2016 Long-term epidemiological survey of *Kudoa thyrsites* (Myxozoa) in Atlantic salmon (*Salmo salar* L.) from commercial aquaculture farms. *J. Fish Dis.* **39**, 929–946.
- Robinson CV, Uren Webster TM, Cable J, James J, Consuegra S. 2018 Simultaneous detection of invasive signal crayfish, endangered white-clawed crayfish and the crayfish plague pathogen using environmental DNA. *Biol. Conserv.* **222**, 241–252.
- Sassoubre LM, Yamahara KM, Gardner LD, Block BA, Boehm AB. 2016 Quantification of Environmental DNA (eDNA) Shedding and Decay Rates for Three Marine Fish. *Environ. Sci. Technol.* **50**, 10456–10464.
- Sato Y, Mizuyama M, Sato M, Minamoto T, Kimura R, Toma C. 2019 Environmental DNA metabarcoding to detect pathogenic *Leptospira* and associated organisms in leptospirosis-endemic areas of Japan. *Sci. Rep.* **9**, 1–11.